# Double burden of gestational diabetes and pregnancy-induced hypertension in Ethiopia: A systematic review and meta-analysis of observational studies

Eyob Girma Abera[1,2☯]*, Esayas Kebede Gudina[2,3☯], Ermias Habte Gebremichael[3], Demisew Amenu Sori[4], Daniel Yilma[2,3☯]

1 Department of Public Health, Jimma University, Jimma, Oromia, Ethiopia, 2 Jimma University Clinical Trial Unit, Jimma, Oromia, Ethiopia, 3 Department of Internal Medicine, Jimma University, Jimma, Oromia, Ethiopia, 4 Department of Obstetrics and Gynecology, Jimma University, Jimma, Ethiopia

☯ These authors contributed equally to this work.
* eyob.girma@ju.edu.et, eyobgirma840@gmail.com

**Data Availability Statement:** All relevant data are within the manuscript.

## Abstract

### Background

The coexistence of gestational diabetes mellitus (GDM) and pregnancy-induced hypertension (PIH) amplifies the risk of maternal and perinatal mortality and complications, leading to more severe adverse pregnancy outcomes. This systematic review and meta-analysis aimed to assess the double burden of GDM and PIH (GDM/PIH) among pregnant women in Ethiopia.

### Methods

A comprehensive systematic search was conducted in the databases of PubMed, Cochrane Library, Science Direct, Embase, and Google Scholar, covering studies published up to May 14, 2023. The analysis was carried out using JBI SUMARI and STATA version 17. Subgroup analyses were computed to demonstrate heterogeneity. A sensitivity analysis was performed to examine the impact of a single study on the overall estimate. Publication bias was assessed through inspection of the funnel plot and statistically using Egger's regression test.

### Result

Of 168 retrieved studies, 15 with a total of 6391 participants were deemed eligible. The pooled prevalence of GDM/PIH co-occurrence among pregnant women in Ethiopia was 3.76% (95% CI; 3.29–4.24). No publication bias was reported, and sensitivity analysis suggested that excluded studies did not significantly alter the pooled prevalence of GDM/PIH co-occurrence. A statistically significant association between GDM and PIH was observed, with pregnant women with GDM being three times more likely to develop PIH compared to those without GDM (OR = 3.44; 95% CI; 2.15–5.53).

**Funding:** The author(s) received no specific funding for this work.

**Competing interests:** The authors have declared that no competing interests exist.

## Conclusion

This systematic review and meta-analysis revealed a high dual burden of GDM and PIH among pregnant women in Ethiopia, with a significant association between the two morbidities. These findings emphasize the critical need for comprehensive antenatal care programs in Ethiopia to adequately address and monitor both GDM and PIH for improved maternal and perinatal health outcomes.

## Introduction

Gestational diabetes mellitus (GDM) is defined as any degree of glucose intolerance that occurs during pregnancy. It is characterized by high blood glucose levels that were either absent or well-controlled before pregnancy [1]. Pregnancy-induced hypertension (PIH) is defined as new hypertension (blood pressure $\geq$ 140/90mmHg) that appears at 20-weeks or later in gestational age, with or without proteinuria ($\geq$ 300mg/24 hours). This category includes gestational hypertension (GH), pre-eclampsia, and eclampsia [2,3]. Although each of these conditions can impact pregnancy outcomes on its own, the combination of GDM and PIH increases the risk of preterm labor, stillbirth, macrosomia, low birth weight, intrauterine growth restriction (IUGR), perinatal death, and maternal death [4,5].

The prevalence of GDM has increased two to threefold in the past decade, driven by increasing obesity rates and lifestyle changes [6]. According to the International Diabetes Federation (IDF) estimates, the overall global prevalence of diabetes in pregnancy was 15.5% in 2019, with GDM accounting for 12.8% [7]. However, by 2021, the GDM prevalence escalated to 14.7% [8]. On the other hand, PIH affects 5–10% of all pregnancies worldwide [9], with incidence increasing from 16.3 million to 18.08 million between 1990 and 2019 [10]. In Ethiopia, PIH burden ranges from 2.23 to 23.42% according various studies [11–17], while a systematic review and meta-analysis indicated a GDM prevalence of 12.04% [18].

Most studies indicate that GDM is independently associated with the occurrence of PIH. After adjustment for maternal age and body mass index (BMI), GDM increases the risk of PIH by approximately 1.5-fold [4]. Besides, GDM is a significant risk factor for recurrent and new postnatal PIH in the absence of a prior PIH history [6]. The history of GDM in first pregnancy also increases the risk of PIH in subsequent pregnancies [4,6]. Moreover, women with a history of GDM and PIH are at risk for development of chronic hypertension later in life [5]. Furthermore, these women face an increased risk of future maternal diabetes and cardiovascular events due to oxidative stress, release of pro-inflammatory factors, and vascular endothelial dysfunction [4–6]. This suggests an association between these conditions. However, despite the existence of some studies, there remains a lack of comprehensive evidence regarding the precise magnitude and correlation of PIH/GDM co-occurrence among pregnant women in Ethiopia. Therefore, this systematic review and meta-analysis aimed to assess the prevalence of GDM and PIH co-occurrence and their correlation among this population.

## Methods

The protocol was registered in the International Prospective Register of Systematic Reviews (PROSPERO) with registration number (CRD42023460613). This systematic review was conducted in accordance with the Joanna Briggs Institute (JBI) methodology for systematic reviews of prevalence and incidence, and risk and etiology review [19] with the updated guideline of the Preferred Reporting Items for Systematic Reviews and Meta-Analysis (PRISMA 2020) (S1 Table) [20].

## Search strategy

Observational studies that assessed the prevalence of GDM/PIH co-occurrence among pregnant women in Ethiopia were included in this review. All relevant articles published up to May 14, 2023, were included in this review with no language restrictions. The studies were identified through comprehensive searches of databases including PubMed, Cochrane Library, Science Direct, Embase, and Google Scholar. Medical Subject Headings (MeSH) terms for "Pregnancy-Induced Hypertension," "Gestational Diabetes Mellitus," "Pregnant Women," and "Ethiopia" were used, along with associated keywords and phrases. These terms were applied both independently and in combination, using Boolean operators 'OR' and 'AND' to systematically search for eligible articles. Additionally, the reference lists of included studies were reviewed to identify further relevant articles. The search was conducted across all specified electronic databases from March 15 to May 14, 2023 (S1 Appendix). Subsequently, all identified studies from the databases were listed along with the reasons for exclusion (S2 Appendix).

## Eligibility criteria

The inclusion criteria for this review and meta-analysis comprised studies involving pregnant women diagnosed with both GDM and PIH in Ethiopia. Articles with accessible full-text and observational studies were included. However, studies with duplicate publications or incomplete outcome data were excluded.

## Study selection

Retrieved and identified citations were collated and exported into EndNote version X9.3.3 (Clarivate Analytics, PA, USA) after removing duplicate publications. Titles and abstracts were screened by three independent reviewers (EGA, EKG, and DY) to assess against the inclusion criteria. Potentially relevant studies were then retrieved in full and their citation details were imported into the Joanna Briggs Institute System for the Unified Management, Assessment and Review of Information (JBI SUMARI) (JBI, Adelaide, Australia) [21]. The full text of selected citations was thoroughly assessed against the inclusion criteria by three independent reviewers (EGA, EKG, and DY). Any disagreements that arose between the reviewers at each stage of the selection process were resolved through discussion.

## Outcome measurements

There were two main outcomes in this systematic review and meta-analysis. The primary outcome was the pooled prevalence of GDM/PIH co-occurrence among pregnant women in Ethiopia. The secondary outcome was to assess the relationship between GDM and PIH, as well as to estimate the pooled effect size of the risk of GDM on the occurrence of PIH among pregnant women in Ethiopia.

Studies reporting outcomes related to GDM as well as studies reporting outcomes related to PIH, including its subtypes such as—eclampsia, pre-eclampsia, and gestational hypertension alone or in combination, were categorized as studies on PIH for the purpose of this review. Pregnant women with pre-existing DM and hypertension were not considered outcomes of interest.

## Assessment of methodological quality

Eligible studies were critically appraised by three independent reviewers (EGA, EKG, and DY) for methodological quality using a standardized critical appraisal instrument developed by JBI

for systematic reviews of observational studies between May 18 and May 24, 2023 (S2 Table) [22]. Any disagreements that arose between the reviewers at each stage of the selection process were resolved through discussion.

## Data extraction

Data were extracted by three independent reviewers (EGA, EKG, and DY) using the standardized data extraction tool developed by JBI for systematic reviews of observational studies between May 25 and June 03, 2023 (S3 Table) [23]. The extracted data included specific details about the study design, setting/context, and year/ timeframe for data collection, participant characteristics (study inclusion/exclusion information), condition and measurement method, and description of the main results. Any disagreements that arose between the reviewers were resolved through discussion.

## Data synthesis and analysis

The extracted data were pooled in statistical proportional meta-analysis using JBI SUMARI and STATA version 17. Effect sizes were expressed as a proportion with 95% confidence intervals around the summary estimate. Adjusted odds ratios (AOR) with their upper and lower bounds were extracted for significant variables. The mean and standard deviation of the studies were pooled. Statistical analyses were performed using the Freeman-Tukey transformation with a random effects model. Heterogeneity was assessed using $I^2$ and its corresponding P-value. Heterogeneity levels were categorized as low ($I^2 < 25\%$), moderate ($25\% \leq I^2 \leq 50\%$), or high ($I^2 > 50\%$) [24]. To decide the source of heterogeneity, a subgroup analysis was performed by region, sample size and study design. To assess potential missing data, publication bias was evaluated through visual inspection of the funnel plot and statistically using Egger's regression test. Besides, sensitivity analysis was performed to examine the influence of a single study on the overall estimate.

# Result

From a total of 168 retrieved studies, 15 were found eligible and included in the final analysis (Fig 1).

## Characteristics of included studies

The mean age of the participants in each study ranged from 25.6 (±4.8) [25] to 33.53 (±5.83) [26] years, with pooled mean age of 27.62 (±5.35) years. The highest prevalence of GDM/PIH co-occurrence among pregnant women was reported equally in Oromia Region and Addis Ababa City (13.58%) [27,28]. The sample size of the studies ranged from 128 [29] to 1834 [25] (Table 1).

## Pooled prevalence of GDM/PIH co-occurrence

From a total of 15 studies with 6391 study participants, the pooled prevalence of GDM/PIH co-occurrence among pregnant women in Ethiopia was 3.76% (95% CI; 3.29–4.24) with significant statistical heterogeneity ($I^2$ = 91.2%, p = <0.0001) (Fig 2).

## Subgroup analysis by region

The highest heterogeneity, ($I^2$ = 96.6%, p = <0.001), with pooled prevalence of 5.35% (95% CI; 4.51–6.18) was observed in the Oromia Region from four studies. Conversely, with an equal number of studies (n = 3) and low heterogeneity, the southern nations, nationalities and

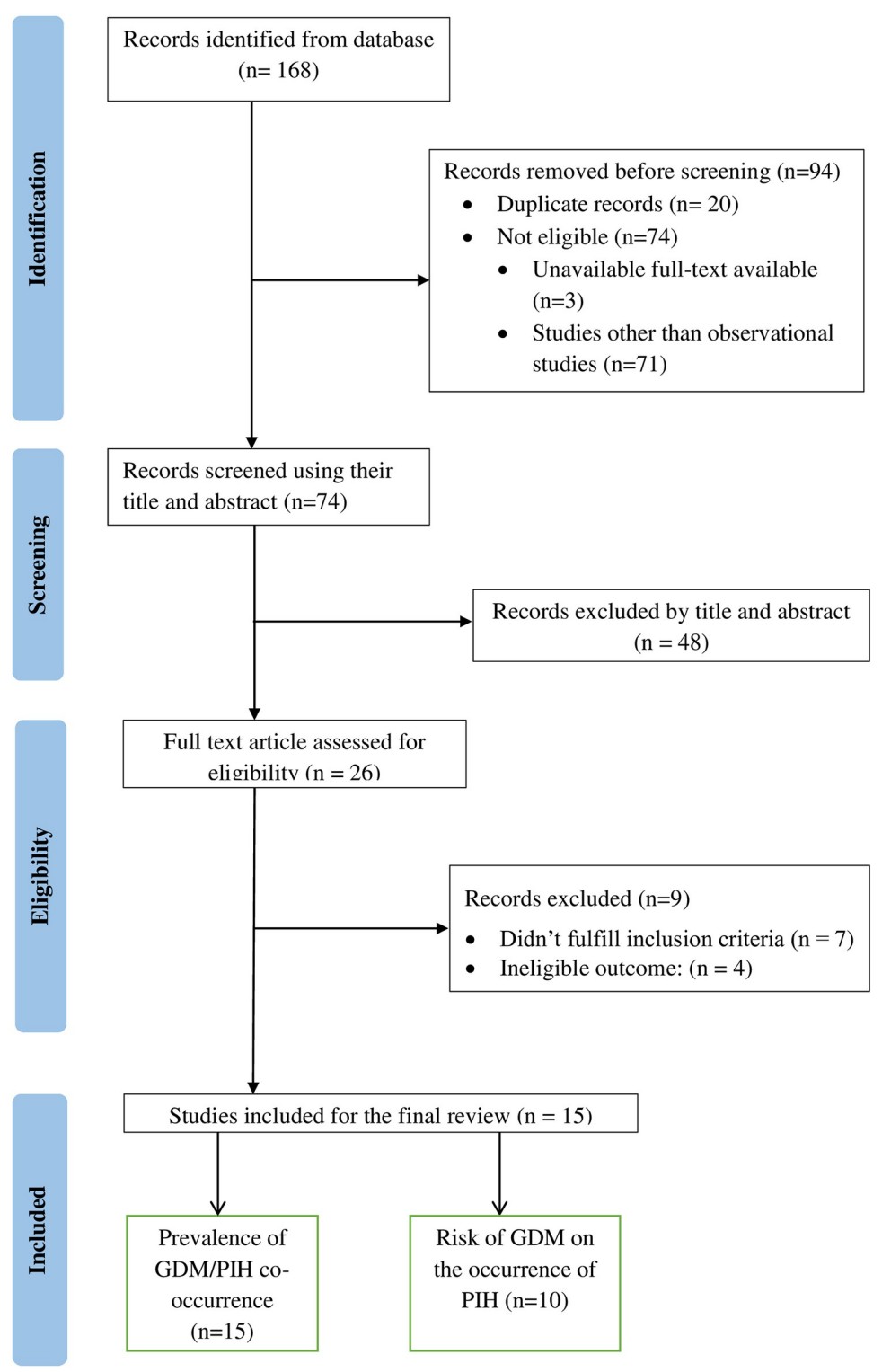

**Fig 1. PRISMA flow diagram of search and study selection process.**

**Table 1. Key characteristics of studies included in this systematic review and meta-analysis.**

| Study | Region | Study design | Sample | Mean age(SD) | Prevalence (%) |
|---|---|---|---|---|---|
| Andarge 2020 [30] | SNNPR | Cross-Sectional | 242 | 27.70 (4.00) | 2.07 |
| Ayalew 2019 [31] | Amhara | Cross-Sectional | 193 | 27.73 (4.30) | 3.63 |
| Duko 2021 [32] | SNNPR | Case Control | 283 | 26.10 (5.40) | 3.53 |
| Firisa 2021 [33] | Addis Ababa | Cross-Sectional | 297 | 28.98 (9.85) | 5.72 |
| Haymanot 2020 [34] | Amhara | Case Control | 200 | 29.07 (6.39) | 4.00 |
| Kahsay 2018 [35] | Tigray | Case Control | 330 | 26.94 (5.73) | 2.12 |
| Katore 2021 [36] | Oromia | Case Control | 302 | 26.29 (4.70) | 0.99 |
| Kidane 2022 [37] | Oromia | Case Control | 312 | 29.08 (6.42) | 6.41 |
| Welesemayat 2020 [26] | Tigray | Cohort | 476 | 33.53 (5.83) | 2.10 |
| Boka 2019 [27] | Oromia | Cross-Sectional | 346 | 26.10 (5.40) | 13.58 |
| Debele 2023 [29] | Addis Ababa | Case Control | 128 | 29.41 (4.84) | 9.38 |
| Eshetu 2019 [28] | Addis Ababa | Cross-Sectional | 346 | 30.80 (4.70) | 13.58 |
| Muche 2020 [38] | Amhara | Cohort | 694 | 27.67 (5.03) | 2.16 |
| Wakwoya 2018 [25] | Oromia | Case Control | 1834 | 25.60 (4.80) | 8.51 |
| Wolka 2022 [39] | SNNPR | Cohort | 408 | 27.54 (4.85) | 3.19 |

SNNPR, Southern nations, nationalities and peoples; SD, standard deviation.

peoples (SNNPR) and Amhara regions had the lowest and relatively similar pooled prevalence, 2.9% (95% CI; 1.78–4.02), ($I^2 = 0.0\%$, p = 0.569) and 2.56% (95% CI; 1.59–3.53), ($I^2 = 3.3\%$, p = 0.356), respectively (S1 Fig).

## Subgroup analysis by sample size

Despite the similar pooled prevalence rates, six studies with a sample size of <300 exhibited moderate heterogeneity ($I^2 = 48.6\%$, p = <0.083) compared to the high heterogeneity of nine studies with a sample size of ≥300, ($I^2 = 94.7\%$, p = <0.0001) (S2 Fig).

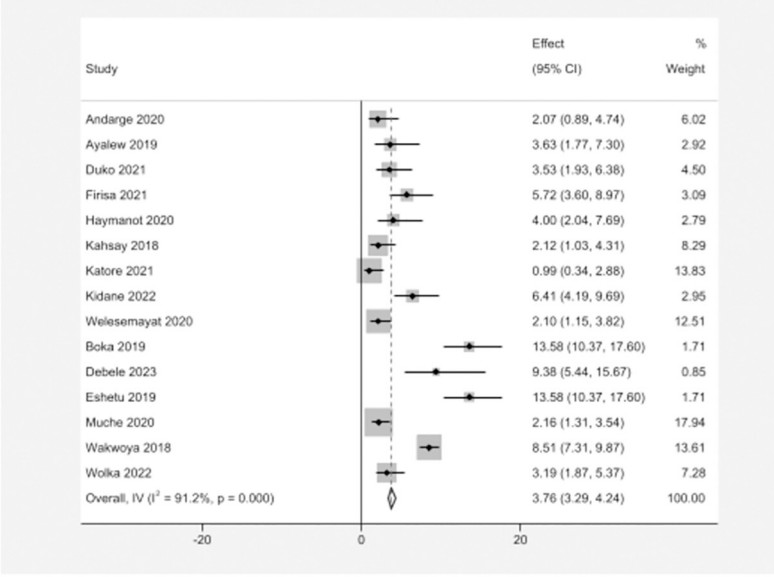

**Fig 2. Pooled prevalence of GDM/PIH co-occurrence among pregnant women in Ethiopia.**

## Subgroup analysis by study design

Three cohort studies reported a lower pooled prevalence of 2.34% (95% CI; 1.57–3.11) with no heterogeneity ($I^2$ = 0.0%, p = 0.568) compared to cross-sectional and case-control studies (S3 Fig).

## Publication bias and sensitivity analysis

A sensitivity analysis was conducted to evaluate the impact of each study on the pooled estimated prevalence of GDM/PIH co-occurrence among pregnant women in Ethiopia. This analysis was performed both for all included studies and specifically by excluding case-control studies, using a step-by-step exclusion process with a random effects model. The results indicated that the exclusion of individual studies, whether across all study designs (S4 Fig) or specifically excluding case-control studies (S5 Fig), did not significantly alter the pooled prevalence.

The included studies were assessed for a potential publication bias using a funnel plot (S6 Fig) and Egger's test that indicated the absence of a publication bias as *P*-values >0.05, suggesting no small study effect. Furthermore, there were no missing data reported in the included studies, so no imputation methods were required (S4 Table).

## Risk of GDM on the occurrence of PIH

Ten studies have reported an association between GDM and PIH [25,26,29–35,37]. Except for two studies [29,32], eight studies found a significant association between GDM and PIH. However, four studies [26,29,33,34] only reported the association without providing the odds of occurrence (OR). Six studies [25,30–32,35,37] involving 3194 pregnant women evaluated the risk of GDM on the occurrence of PIH, revealing 205 cases of GDM among women diagnosed with PIH (Table 2).

Using a random-effects model, the pooled analysis revealed a statistically significant association between GDM and PIH with insignificant heterogeneity ($I^2$ = 19%, p = 0.29). The odds for the occurrence of PIH were three times higher in pregnant women with GDM compared to women without GDM [OR = 3.44; (95% CI; 2.15–5.53)] (Fig 3).

**Table 2. Characteristics of studies that reported an association between GDM and PIH.**

| Study | Region | Study design | Sample | Prevalence (%) |
|---|---|---|---|---|
| Andarge 2020 [30][a, b] | SNNPR | Cross-Sectional | 242 | 2.07 |
| Ayalew 2019 [31] [a, b] | Amhara | Cross-Sectional | 193 | 3.63 |
| Duko 2021 [32][b] | SNNPR | Case Control | 283 | 3.53 |
| Kahsay 2018 [35][a, b] | Tigray | Case Control | 330 | 2.12 |
| Kidane 2022 [37][a, b] | Oromia | Case Control | 312 | 6.41 |
| Wakwoya 2018 [25][a, b] | Oromia | Case Control | 1834 | 8.51 |
| Firisa 2021 [33][a, c] | Addis Ababa | Cross-Sectional | 297 | 5.72 |
| Haymanot 2020 [34][a, c] | Amhara | Case Control | 200 | 4.00 |
| Welesemayat 2020 [26][a, c] | Tigray | Cohort | 476 | 2.10 |
| Debele 2023 [29][c] | Addis Ababa | Case Control | 128 | 9.38 |

[a]Studies reported a significant association between GDM and PIH, (p-value≤0.05), (n = 8).

[b]Studies designed to assess the risk of GDM on the occurrence of PIH, (n = 6).

[c]Studies reported only the association without reporting the OR, (n = 4).

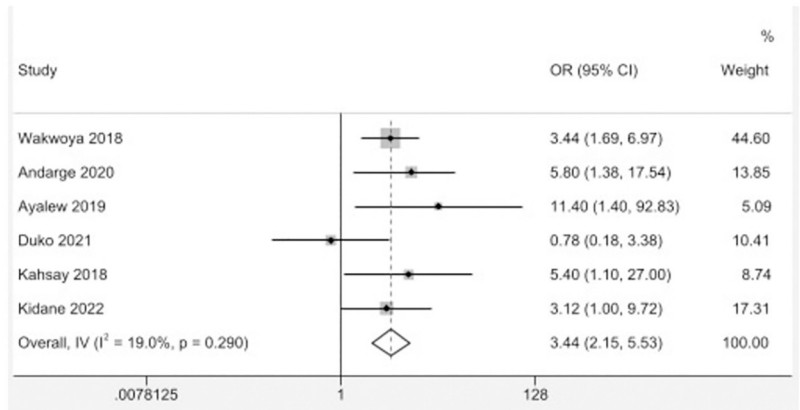

**Fig 3. The risk of GDM on the occurrence of PIH among pregnant women in Ethiopia.**

## Discussion

This systematic review and meta-analysis, based on 15 studies involving 6391 pregnant women in Ethiopia, evaluated the prevalence and co-occurrence of GDM and PIH. Notably, all studies were published since 2018, despite no restrictions on publication year.

The pooled prevalence of GDM/PIH co-occurrence among pregnant women in Ethiopia was 3.76% (95% CI; 3.29–4.24). A comparable finding was observed in Iran, where a study involving 615 pregnant women reported a GDM/PIH prevalence of 3.18% (95% CI: 1.13–8.94) [40]. However, the current finding is higher than a cohort study conducted in Taiwan among 2,297,613 pregnant women who gave birth between 2004 and 2015 using data from the Taiwan National Health Insurance Research Database (TNHIRD) that reported a 0.03% prevalence rate of GDM/PIH co-occurrence [41]. Similarly, the finding of this systematic review and meta-analysis is higher than a population-based case-control study conducted in Washington State among 62,982 pregnant women using 1992–1998 Washington State birth certificate and hospital discharge records, where the prevalence of GDM/PIH co-occurrence was 1.1% [42]. Lower prevalence rates of GDM/PIH co-occurrence were also reported from Sweden (0.14% among 10,659 pregnant women) [43] and Latin America and the Caribbean (0.1% among 878,680 pregnant women) [44].

These differences in prevalence could be attributed to variations in healthcare systems, utilization of antenatal care services, and racial disparities. Notably, in previous study, black women with GDM were almost twice as likely to develop PIH compared to other races [45]. Moreover, our study's findings, based on a smaller study population, may differ from those of large cohort studies. The type of study design can also significantly impact the reported prevalence of GDM and PIH co-occurrence. Population-based studies are generally preferable for assessing prevalence as they provide a more accurate reflection of the burden in the general population, unlike hospital-based studies that often represent high-risk populations [46,47]. This distinction is crucial for understanding the actual burden of these conditions.

However, our results are lower than a cohort study conducted in Ontario, Canada, which reported a prevalence rate of 6.29% for GDM/PIH co-occurrence among 270,843 pregnant women who gave birth between 2012 and 2016 [48]. Additionally, studies conducted in Taiwan [49] and India [50] reported higher prevalence rates of 6.67% and 4.76%, respectively. This disparity may be attributed to differences in lifestyle management, including diet and physical exercise, as well as the distribution of non-communicable diseases over time, despite having better healthcare systems compared to Ethiopia.

The odds for the occurrence of PIH were three times higher in pregnant women with GDM than women without [OR = 3.44; (95% CI; 2.15–5.53)]. This finding aligns with a cohort study conducted in Sweden among 10,659 pregnant women, which reported a significantly increased risk of PIH among those with GDM (OR = 3.16, 95 percent CI: 1.65, 6.03) compared to those without [43]. Similarly, a retrospective study in Germany involving 647,392 pregnancies found that the odds of PIH had increased among women with GDM (OR = 1.29; 95% CI = 1.19, 1.41) [51]. Another cohort study in Germany also reported a significant associated between GDM and a higher risk of preeclampsia PIH (OR = 1.9; 95% CI = 1.7, 2.1) [52]. In addition, a study in Sweden among 430,852 pregnant women reported GDM to be significantly associated with a higher risk of PIH (OR = 1.61; 95% CI = 1.39, 1.86) [53].

A French study of 15 maternity units also found that the odds of PIH increases five times among pregnant women with GDM compared to those without (OR = 2.86, 95% CI: 1.05, 7.83) [54]. However, a case-control study in the US reported a different outcome. Although a 1.5-fold increased risk of severe and mild preeclampsia and a 1.4-fold increase in gestational hypertension was reported, no significant association was observed between GDM and PIH [AOR = 1.27, 95% CI: (0.52, 3.15)] after adjustment for BMI, age, ethnicity, parity, and adequacy of prenatal care [42].

The variability in the association between GDM and PIH across studies may stem from differences in diagnostic criteria, lifestyle factors, and genetic predispositions. Variations in the thresholds and timing for diagnosing GDM and PIH could influence reported associations. Additionally, lifestyle factors such as diet and physical activity, as well as regional healthcare practices, may influence the strength of the association. Genetic and ethnic factors could also impact the likelihood of co-occurrence [55]. Furthermore, confounding variables like obesity and hypertension, which are not always adequately controlled for, may affect the observed relationship between GDM and PIH [56].

The co-occurrence of GDM and PIH has been documented across various settings, yet the exact mechanisms behind this association are still not fully understood [55,57]. Both conditions share common risk factors, including insulin resistance and endothelial dysfunction, which may contribute to their simultaneous presence [57]. GDM-induced metabolic disturbances, such as hyperglycemia and oxidative stress, can exacerbate PIH by impairing vascular function and promoting inflammation [57]. Elevated levels of advanced glycation end-products and pro-inflammatory cytokines like tumor necrosis factor-$\alpha$ and interleukin-6 in GDM further contribute to endothelial dysfunction and the development of PIH [6,58]. Additionally, GDM can lead to excessive activation of neutrophils and the release of neutrophil extracellular traps, which impair placental blood flow and increase the risk of preeclampsia [59,60].

For clinicians, these findings highlight the need for routine screening of GDM and PIH due to their high co-occurrence and potential complications. Public health experts and policymakers should focus on developing targeted interventions and allocating resources to improve maternal health services. Researchers should investigate the mechanisms behind these prevalence rates, while pregnant women should be aware of the importance of early diagnosis and management to reduce associated risks.

## Strength and limitation

This systematic review represents the first comprehensive assessment of the associations between GDM and PIH in Ethiopia, offering valuable insights into the prevalence and relationships between these conditions in the Ethiopian context. The methodological quality of this review was strengthened by involving three reviewers, employing a comprehensive search strategy, and adhering to JBI critical appraisal instruments and PRISMA guidelines.

However, there are some limitations to consider. The review encompassed a variety of observational study designs, including cross-sectional, cohort, and case-control studies. Each design has inherent limitations. The exclusion of unavailable full-text articles may have affected the accuracy of pooled prevalence and associations, potentially introducing selection bias. Additionally, the variability in diagnostic criteria across primary studies—such as differences between diagnosed and undiagnosed GDM or PIH—could affect the reported estimates and limit the comparability of findings. Another limitation of this review is the pooling of prevalence data despite high heterogeneity among the included studies.

## Conclusion

This review found a significant association between GDM and PIH among pregnant women in Ethiopia. GDM-diagnosed women were three times more likely to develop PIH compared to those without it. This highlights the importance of routine screening for GDM and PIH early in pregnancy for all pregnant women. Regular monitoring of blood glucose levels and blood pressure throughout pregnancy is essential to track changes and identify any emerging issues promptly. Effective interventions are also needed to improve maternal health outcomes and reduce healthcare burden, while further research is warranted to understand underlying factors and evaluate intervention strategies.

## Supporting information

**S1 Table. Prisma 2020 checklist for reporting the findings of the systematic review and meta-analysis on the double burden of gestational diabetes and pregnancy-induced hypertension in Ethiopia.**
(PDF)

**S2 Table. Critical appraisal of studies in the systematic review and meta-analysis on the double burden of gestational diabetes and pregnancy-induced hypertension in Ethiopia.**
(PDF)

**S3 Table. Extracted data of included studies in the systematic review and meta-analysis on the double burden of gestational diabetes and pregnancy-induced hypertension in Ethiopia.**
(PDF)

**S4 Table. Egger's test of prevalence of GDM/PIH co-occurrence among pregnant women in Ethiopia.**
(PDF)

**S1 Fig. Subgroup analysis of pooled prevalence of GDM/PIH co-occurrence among pregnant women in Ethiopia by region.**
(TIF)

**S2 Fig. Subgroup analysis of pooled prevalence of GDM/PIH co-occurrence among pregnant women in Ethiopia by sample size.**
(TIF)

**S3 Fig. Subgroup analysis of pooled prevalence of GDM/PIH co-occurrence among pregnant women in Ethiopia by study design.**
(TIF)

**S4 Fig. Sensitivity analysis of the prevalence of GDM/PIH co-occurrence among pregnant women in Ethiopia across all study designs.**
(TIF)

**S5 Fig. Sensitivity analysis of the prevalence of GDM/PIH co-occurrence among pregnant women in Ethiopia, excluding case-control studies.**
(TIF)

**S6 Fig. Funnel plot of pooled prevalence of GDM/PIH co-occurrence among pregnant women in Ethiopia.**
(TIF)

**S1 Appendix. Search strategy of the systematic review and meta-analysis on the double burden of gestational diabetes and pregnancy-induced hypertension in Ethiopia.**
(PDF)

**S2 Appendix. List of all studies identified in the literature search for the systematic review and meta-analysis on the double burden of gestational diabetes and pregnancy-induced hypertension in Ethiopia.**
(PDF)

## Acknowledgments

We would like to thank all authors of studies included in this systematic review and meta-analysis.

## Author Contributions

**Conceptualization:** Eyob Girma Abera, Ermias Habte Gebremichael.

**Data curation:** Eyob Girma Abera.

**Formal analysis:** Eyob Girma Abera.

**Methodology:** Eyob Girma Abera.

**Project administration:** Daniel Yilma.

**Software:** Eyob Girma Abera.

**Supervision:** Esayas Kebede Gudina, Daniel Yilma.

**Validation:** Eyob Girma Abera, Esayas Kebede Gudina, Ermias Habte Gebremichael, Demisew Amenu Sori, Daniel Yilma.

**Visualization:** Esayas Kebede Gudina, Demisew Amenu Sori.

**Writing – original draft:** Eyob Girma Abera.

**Writing – review & editing:** Eyob Girma Abera, Esayas Kebede Gudina, Ermias Habte Gebremichael, Demisew Amenu Sori, Daniel Yilma.

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
