## [Decision Letter · Decision Letter 0]

5 Aug 2024

PONE-D-24-13046Double burden of pregnancy-induced hypertension and gestational diabetes mellitus among pregnant women in Ethiopia: a systematic review and meta-analysisPLOS ONE

Dear Dr. Abera,

Thank you for submitting your manuscript to PLOS ONE. After careful consideration, we feel that it has merit but does not fully meet PLOS ONE’s publication criteria as it currently stands. Therefore, we invite you to submit a revised version of the manuscript that addresses the points raised during the review process.

We look forward to receiving your revised manuscript.

Kind regards,

Berhanu Elfu Feleke, PhD Fellow

Academic Editor

PLOS ONE

2. We note that your Data Availability Statement is currently as follows: [All relevant data are within the manuscript.]

Additional Editor Comments:

• Format tables to the publication standards.

• How did you extract the prevalence data from case-control studies?

• Can you provide the age-specific estimates?

• Discuss the significance and implications of your findings to clinicians, public health experts, managers, decision/policy makers, researchers, and to pregnant women.

• Comment on the effects of different diagnostic criteria (diagnosed/undiagnosed GDM or PIH) across primary studies to your estimate.

• Have a descriptive title to all figures and tables.

Reviewers' comments:

Reviewer's Responses to Questions

**Comments to the Author**

1. Is the manuscript technically sound, and do the data support the conclusions?

Reviewer #1: Yes

Reviewer #2: No

2. Has the statistical analysis been performed appropriately and rigorously? 

Reviewer #1: Yes

Reviewer #2: No

3. Have the authors made all data underlying the findings in their manuscript fully available?

Reviewer #1: Yes

Reviewer #2: No

4. Is the manuscript presented in an intelligible fashion and written in standard English?

Reviewer #1: Yes

Reviewer #2: Yes

5. Review Comments to the Author

Reviewer #1: Thank you for inviting me to review the manuscript on Double burden of pregnancy-induced hypertension and gestational diabetes mellitus among pregnant women in Ethiopia: a systematic review and meta-analysis. I think the title is very impressive and gives a good clue for scholars and policy makers. In general the research topic is interesting and it is researchable. I have some comments and suggestions for the author.

1. Title should be smart and should answer what, whom, when and what questions ; but your topic not answer the question when?

2. On abstract method section the period of study should be specified

Introduction

1. Line 72 you side that, there is limited evidence regarding the magnitude and correlation of PIH/GDM co-occurrence among pregnant women in Ethiopia; but there is around 15 original research evidences based on your reports; how did you see these controversy ideas?

Method section

Line 88 on Search strategy: you reported that you have used different search engines like PubMed, Cochrane Library, Science 88 Direct, Embase, HINARI, and Google Scholar; it is good to use such searching mechanisms ; to be clear you should report in each search engines terms, MUSH terms including the date of search and total number of result in each search clearly by table.

Discussion

Your discussion section is somewhat shallow; you should more explore the co-occurrence of GDM and PIH evidences, how the relations hypothesis from different evidences and then put your hypothesis.

Recommendations

1. You should put recommendations separately to the conclusion

2. You should recommend specifically for the concerned bodies

3. The recommendation should be result based, specific, measurable and achievable

Reviewer #2: This study aims to assess the pregnancy-induced hypertension and gestational diabetes mellitus

among pregnant women in Ethiopia.

What was the definition of GDM and PIH in this review? Were they consistent across studies?

Can the authors please explain how they obtained the data in Table 1. For example, how do they calculate prevalence from a case control study? Furthermore, for many of the papers, the data used to populate Table 1 were not immediately evident in the references.

Add study nos to the references in Table 1

some of references. for example, Duko (is this Belayhun?), Wolka, Katore and Muche are not in reference list

use decimals consistently

Did the authors distinguish between gestational diabetes and type1/type 2 diabetes?

In the introduction, BMI is used without being written out

On page 7, not sure what 25 and 26 in brackets refer to? Also, 29 and 25 in the last line

Prevalence could be affected by the diagnostic criteria used, setting (high risk (hospital) or clinic). Hospitals represent high risk populations, therefore, is not reflective of the burden in Ethiopia? This could also explain the discrepancies with other studies in the discussion. Population-based studies are preferable to assess prevalence, the authors need to include the importance of study designs in the discussion.

Do not use it's rather write out as it is

For the subgroup analysis by sample size, why was 300 selected?

Comment on the quality of the evidence and pooled estimate in the context of the limitations of study designs and reporting, and overall heterogeneity.

How did this review limit publication bias?

Elaborate on the limitations of the systematic review with regard to study design, included studies, etc.

6. PLOS authors have the option to publish the peer review history of their article (what does this mean?). If published, this will include your full peer review and any attached files.

Reviewer #1: **Yes: **Fentahun Yenealem Beyene

Reviewer #2: No

---

## [Author Response · Author response to Decision Letter 0]

14 Aug 2024

Additional Editor Comments

1. Format tables to the publication standards.

• Authors’ response: Thank you for your feedback. We have formatted the tables to meet PLoS publication standards.

2. How did you extract the prevalence data from case-control studies?

• Authors’ response: Thank you for your insightful comment. The concern raised about pooling prevalence from such studies is valid, as it requires careful consideration to avoid misrepresenting the true burden of the conditions. To address this, we carefully aimed to ensure accurate representation. Excluding case-control studies would have limited our understanding of the overall burden. Therefore, we included prevalence data from these studies by recalculating the co-occurrence of GDM and PIH based on the entire population of the study, not just the case group. For example, in the study by Duko et al. (2021), with 283 participants (95 with PIH and 188 without), the prevalence of GDM among the PIH group was reported as 10.5%. To accurately reflect the burden of GDM/PIH co-occurrence, we recalculated the prevalence using the total sample size, resulting in a prevalence of 3.35%.

• Similarly, in the study by Haymanot et al. (2020), out of 200 participants (100 with PIH and 100 without), 8% of those with PIH had GDM. Again, by recalculating based on the total population, the co-occurrence rate was 4%. This method allowed us to provide a more comprehensive and accurate estimation of the double burden of GDM and PIH among pregnant women in Ethiopia.

3. Can you provide the age-specific estimates?

• Authors’ response: Thank you for your valuable feedback. We understand the importance of providing age-specific estimates for PIH/GDM co-occurrence. However, the available data from the included studies allowed us to report only the mean (SD) of age, which summarized the overall age distribution of the study populations. Unfortunately, due to the nature of the data from the included studies and the fact that they did not calculate the prevalence of PIH/GDM within different age groups, we are unable to stratify the prevalence estimates by specific age groups. 

4. Discuss the significance and implications of your findings to clinicians, public health experts, managers, decision/policy makers, researchers, and to pregnant women.

• Authors’ response: Thank you for your suggestion. We have updated the manuscript in accordance with the given comment.

5. Comment on the effects of different diagnostic criteria (diagnosed/undiagnosed GDM or PIH) across primary studies to your estimate.

• Authors’ response: Thank you for the valuable suggestion. We have incorporated the feedback and updated the manuscript accordingly.

6. Have a descriptive title to all figures and tables.

• Authors’ response: Thank you for your feedback. We have made the adjustments accordingly.

Reviewer #1

1. Title should be smart and should answer what, whom, when and what questions; but your topic not answer the question when?

• Authors’ Response: Thank you for your insightful feedback. While we understand the importance of addressing the "when" aspect in research titles, we believe that including a specific time frame is not essential in this case due to the nature of systematic reviews, which synthesize data across various time periods. Our focus is on the overall prevalence and co-occurrence of diabetes and hypertension in Ethiopia, rather than a specific time frame. However, the "when" question is addressed in the search strategy section, where we included studies published up to May 14. Therefore, we have revised the title to “Double burden of gestational diabetes and hypertension in Ethiopia: A systematic review and meta-analysis of observational studies” to better reflect the study's scope and intent.

2. On abstract method section the period of study should be specified

• Authors’ response: Thank you for your suggestion. We have updated the abstract section accordingly.

3. Introduction: Line 72 you said that, there is limited evidence regarding the magnitude and correlation of PIH/GDM co-occurrence among pregnant women in Ethiopia; but there is around 15 original research evidences based on your reports; how did you see these controversy ideas?

Authors’ response: Thank you for highlighting this point. While it is true that our review included 15 original research studies, the limited evidence we referred to concerns the comprehensive understanding and synthesis of the magnitude and correlation of PIH and GDM co-occurrence in Ethiopia. To clarify this and address any potential confusion, we have updated the statement as follows: "Despite the existence of some studies, there remains a lack of comprehensive evidence regarding the precise magnitude and correlation of PIH/GDM co-occurrence among pregnant women in Ethiopia.”

4. Method section: Line 88 on Search strategy: you reported that you have used different search engines like PubMed, Cochrane Library, Science 88 Direct, Embase, HINARI, and Google Scholar; it is good to use such searching mechanisms; to be clear you should report in each search engines terms, MESH terms including the date of search and total number of result in each search clearly by table.

• Authors’ response: Thank you for your feedback. We previously uploaded the search strategy details; however, we have now adjusted it as per the suggestion. The updated information, including specific search terms, MeSH terms, search dates, and the total number of results for each database, has been uploaded.

5. Discussion: Your discussion section is somewhat shallow; you should more explore the co-occurrence of GDM and PIH evidences, how the relations hypothesis from different evidences and then put your hypothesis.

• Authors’ response: Thank you for your suggestion. We have updated the discussion section accordingly.

6. You should put recommendations separately to the conclusion

• Authors’ response: We highly value the reviewer’s suggestions. To stick with the publication standards, we kept the recommendation statements under the conclusion but as separate sentences.

7. You should recommend specifically for the concerned bodies

• Authors’ response: Thank you for your feedback. Our recommendation statements in the previous version “This highlights the importance of routine screening for GDM and PIH early in pregnancy for all pregnant women (for Healthcare workers). Regular monitoring of blood glucose levels and blood pressure throughout pregnancy is essential to track changes and identify any emerging issues promptly (for Healthcare workers). Effective interventions are also needed to improve maternal health outcomes and reduce healthcare burden (for Public health, managers, and policy makers), while further research is warranted to understand underlying factors and evaluate intervention strategies (for Researchers)” have implicitly addressed the concerned stakeholders..

8. The recommendation should be result based, specific, measurable and achievable

• Authors’ response: Thank you for your feedback. We have updated the recommendations under the conclusion section to ensure they are specific, measurable, and achievable. The revised recommendations now target key stakeholders, including clinicians, public health experts, managers, decision/policy makers, and researchers.

Reviewer #2

1. What was the definition of GDM and PIH in this review? Were they consistent across studies?

• Authors’ response: Thank you for your observation. In this review, the definition of GDM was consistently applied across all included studies as "Any degree of glucose intolerance that occurs during pregnancy, characterized by high blood glucose levels that were either absent or well-controlled before pregnancy." 

• For PIH, the categorization included gestational hypertension (GH), pre-eclampsia, and eclampsia. However, there was variability in how PIH was reported across studies. Some studies used the term PIH broadly, while others specified pre-eclampsia or eclampsia. To ensure comprehensive reporting and avoid underestimating the burden of PIH, we adopted a broad classification approach in our outcome measurements. Studies reporting outcomes related to these conditions, whether alone or in combination, were included under PIH for the purpose of this review. This approach allowed us to include studies with different terminologies and maintain a more inclusive assessment of the prevalence and association of GDM and PIH.

• We have addressed this concern on page 4, lines 117-120, under sub heading “Outcome measurements”.

2. Can the authors please explain how they obtained the data in Table 1? For example, how do they calculate prevalence from a case control study?

• Authors’ response: Thank you for your insightful comment. The concern raised about pooling prevalence from such studies is valid, as it requires careful consideration to avoid misrepresenting the true burden of the conditions.

• To address this, we took a meticulous approach to ensure accurate representation. Excluding case-control studies would have limited our understanding of the overall burden. Therefore, we included prevalence data from these studies by recalculating the co-occurrence of GDM and PIH based on the entire population of the study, not just the case group.

• For example, in the study by Duko et al. (2021), with 283 participants (95 with PIH and 188 without), the prevalence of GDM among the PIH group was reported as 10.5%. To accurately reflect the burden of GDM/PIH co-occurrence, we recalculated the prevalence using the total sample size, resulting in a prevalence of 3.35%.

• Similarly, in the study by Haymanot et al. (2020), out of 200 participants (100 with PIH and 100 without), 8% of those with PIH had GDM. Again, by recalculating based on the total population, the co-occurrence rate was 4%.

• This method allowed us to provide a more comprehensive and accurate estimation of the double burden of GDM and PIH among pregnant women in Ethiopia.

3. Furthermore, for many of the papers, the data used to populate Table 1 were not immediately evident in the references. Add study nos to the references in Table 1

• Authors’ response: Thank you for your insightful comment. We have updated the table accordingly.

4. Some of references. for example, Duko (is this Belayhun?), Wolka, Katore and Muche are not in reference list

• Authors’ response: Thank you for your comment. The reference labeled as "Duko" in Table 1 refers to the author listed in the Belayhun paper. This naming appeared as "Duko" during the exportation process from EndNote to JBI for critical appraisal. 

• We have included all the references in the table as per your suggestion.

5. Use decimals consistently

• Authors’ response: Thank you for your feedback. We have reviewed the manuscript and ensured that decimals are used consistently throughout the document, including in the table.

6. Did the authors distinguish between gestational diabetes and type1/type 2 diabetes?

• Authors’ response: Yes, we clearly distinguished between gestational diabetes mellitus (GDM) and pre-existing type 1 or type 2 diabetes. We outlined this distinction in the outcome measurements section, specifying that pregnant women with pre-existing diabetes or hypertension were excluded from the study. As stated on page 4, lines 119-120, our review focused solely on outcomes related to GDM and PIH, ensuring that only studies pertaining to gestational conditions were included.

7. In the introduction, BMI is used without being written out

• Authors’ response: Thank you for your observation. We have corrected this in the manuscript by writing out "Body Mass Index (BMI)" in full the first time it appears in the introduction.

8. On page 7, not sure what 25 and 26 in brackets refer to? Also, 29 and 25 in the last line

• Authors’ response: Thank you for pointing this out. The numbers in brackets refer to specific studies cited in the reference list. For instance, reference 25 had the lowest mean age of 25.6 (±4.8), while reference 26 had the highest mean age of 33.53 (±5.83). Additionally, reference 29 corresponds to the study with the smallest sample size of 128, whereas reference 25 had the largest sample size of 1,834. The inclusion of reference numbers within the tables will clarify these points in the revised manuscript.

• To avoid confusion, we have changed the citation style from parentheses ( ) to square brackets [ ].

9. Prevalence could be affected by the diagnostic criteria used, setting (high risk (hospital) or clinic). Hospitals represent high risk populations, therefore, is not reflective of the burden in Ethiopia? This could also explain the discrepancies with other studies in the discussion.

• Authors’ response: Thank you for your comment. We have updated the manuscript accordingly to reflect this consideration.

10. Population-based studies are preferable to assess prevalence; the authors need to include the importance of study designs in the discussion.

• Authors’ response: Thank you for the insightful suggestion. We have updated the manuscript accordingly.

11. Do not use it's rather write out as it is

• Authors’ response: Thank you for the comment. We have updated the manuscript accordingly.

12. For the subgroup analysis by sample size, why was 300 selected?

• Authors’ response: Thank you for your comment. The choice of a sample size of 300 for the subgroup analysis was based on established guidelines for ensuring adequate statistical power and reliable results. Research suggests that a minimum sample size of 200 is generally sufficient for robust data analysis (Hoe, 2008; Singh et al., 2016). However, a sample size of 300 is recommended for achieving better stability in factor analysis and is considered "good" according to Comrey and Lee's criteria (Comrey & Lee, 1992; Tabachnick & Fidell, 2013). This threshold helps ensure that our subgroup analysis is sufficiently powered to detect meaningful effects and improve the robustness of our findings.

13. Comment on the quality of the evidence and pooled estimate in the context of the limitations of study designs and reporting, and overall heterogeneity.

• Authors’ response: Thank you for your feedback. We have updated the 'Strengths and Limitations of the Study' section of the manuscript accordingly based on your suggestion.

14. How did this review limit publication bias?

• Authors’ response: Thank you for your comment. To address potential publication bias, we implemented several strategies. We pre-registered our review protocol to ensure transparency and prevent selective reporting. Although we included studies published up to May 14, 2023, we conducted a comprehensive search strategy to identify both published and unpublished studies with no language restrictions. To ensure that no relevant articles were missed, we employed three reviewers for the selection process. Additionally, we adhered to PRISMA guidelines and assessed publication bias using Egger’s test, which indicated no significant bias. These measures were intended to minimize the influence of publication bias on our findings.

15. Elaborate on the limitations of the systematic review with regard to study design, included studies, etc.

• Authors’ response: Thank you for your comment. We have updated the manuscript in accordance with your suggestion.

---

## [Decision Letter · Decision Letter 1]

9 Sep 2024

PONE-D-24-13046R1Double burden of gestational diabetes and pregnancy-induced hypertension in Ethiopia: A systematic review and meta-analysis of observational studiesPLOS ONE

Dear Dr. Abera,

Thank you for submitting your manuscript to PLOS ONE. After careful consideration, we feel that it has merit but does not fully meet PLOS ONE’s publication criteria as it currently stands. Therefore, we invite you to submit a revised version of the manuscript that addresses the points raised during the review process.

We look forward to receiving your revised manuscript.

Kind regards,

Berhanu Elfu Feleke, PhD Fellow

Academic Editor

PLOS ONE

Journal Requirements:

Additional Editor Comments:

I have seen the authors' response to the reviewers' and my comments, and they have addressed most of the issues. However, they should include a sensitivity analysis and comment on their findings by excluding case-control studies.

Reviewers' comments:

Reviewer's Responses to Questions

**Comments to the Author**

1. If the authors have adequately addressed your comments raised in a previous round of review and you feel that this manuscript is now acceptable for publication, you may indicate that here to bypass the “Comments to the Author” section, enter your conflict of interest statement in the “Confidential to Editor” section, and submit your "Accept" recommendation.

Reviewer #1: All comments have been addressed

2. Is the manuscript technically sound, and do the data support the conclusions?

Reviewer #1: Yes

3. Has the statistical analysis been performed appropriately and rigorously? 

Reviewer #1: Yes

4. Have the authors made all data underlying the findings in their manuscript fully available?

Reviewer #1: Yes

5. Is the manuscript presented in an intelligible fashion and written in standard English?

Reviewer #1: Yes

6. Review Comments to the Author

Reviewer #1: The author address all comments and questions the document sounds for publication. Therefore i have no any additional comment.

7. PLOS authors have the option to publish the peer review history of their article (what does this mean?). If published, this will include your full peer review and any attached files.

Reviewer #1: **Yes: **Fentahun Yenealem Beyene

---

## [Author Response · Author response to Decision Letter 1]

9 Sep 2024

We sincerely appreciate the handling editor and reviewers for their time and insightful comments. As requested, we have revised our manuscript in accordance with the comments provided by the handling editor. Please find our responses below.

Journal Requirements: 

• Authors’ response: Thank you for your feedback. We have reviewed and formatted all references according to the journal’s requirements. If any references were missed or found to be inaccurate, we will be happy to update them.

Additional Editor Comments:

I have seen the authors' response to the reviewers' and my comments, and they have addressed most of the issues. However, they should include a sensitivity analysis and comment on their findings by excluding case-control studies.

• Authors’ response: Thank you for the suggestion. Excluding case-control studies did not significantly alter the pooled prevalence of GDM/PIH co-occurrence among pregnant women in Ethiopia. We have updated the manuscript to reflect these results, along with the supporting figure.

---

## [Editor Report · Decision Letter 2]

13 Sep 2024

Double burden of gestational diabetes and pregnancy-induced hypertension in Ethiopia: A systematic review and meta-analysis of observational studies

PONE-D-24-13046R2

Dear Dr. Abera,

We’re pleased to inform you that your manuscript has been judged scientifically suitable for publication and will be formally accepted for publication once it meets all outstanding technical requirements.

Kind regards,

Berhanu Elfu Feleke, PhD Fellow

Academic Editor

PLOS ONE

Additional Editor Comments (optional):

The authors' addressed all my inquiries so that this manuscript can be accepted for publication at its current stand.
---

## [Editor Report · Acceptance letter]

23 Sep 2024

PONE-D-24-13046R2 

PLOS ONE

Dear Dr. Abera, 

I'm pleased to inform you that your manuscript has been deemed suitable for publication in PLOS ONE. Congratulations! Your manuscript is now being handed over to our production team.

Kind regards, 

on behalf of

Mr. Berhanu Elfu Feleke 

Academic Editor

PLOS ONE